# Duodenal Biopsy Audit: Relative Frequency of Diagnoses, Key Words on Request Forms Indicating Severe Pathology, and Potential Diagnoses for Intraepithelial Lymphocytosis, as a Foundation for Developing Artificial Intelligence Diagnostic Approaches

**DOI:** 10.3390/diagnostics15121483

**Published:** 2025-06-11

**Authors:** Vrinda Shenoy, Jessica L. James, Amelia B. Williams-Walker, Nasyen P. R. Madhan Mohan, Kim N. Luu Hoang, Josephine Williams, Florian Jaeckle, Shelley C. Evans, Elizabeth J. Soilleux

**Affiliations:** 1Department of Pathology, University of Cambridge, Tennis Court Road, Cambridge CB2 1QP, Cambridgeshire, UK; vs534@cantab.ac.uk (V.S.); jlj47@cam.ac.uk (J.L.J.); nprm2@cam.ac.uk (N.P.R.M.M.); fj286@cam.ac.uk (F.J.);; 2Human Research Tissue Bank, Cambridge University Hospitals NHS Foundation Trust, Cambridge CB2 0QQ, Cambridgeshire, UK; josephine.williams15@nhs.net; 3Lyzeum Ltd., Salisbury House, Station Road, Cambridge CB1 2LA, Cambridgeshire, UK

**Keywords:** coeliac disease, audit, duodenal biopsy, gluten sensitivity

## Abstract

**Background/Objectives:** Understanding the diagnostic landscape is essential prior to developing artificial intelligence (AI)-based diagnostic strategies for automating the diagnosis of duodenal biopsies. This study aims to (1) determine the frequencies of different diagnoses seen in endoscopic duodenal biopsies in a large, tertiary referral centre; (2) identify key words on histopathology request forms that could indicate that a biopsy may contain a serious pathology and should not be diagnosed by an AI system; and (3) investigate the proportion of cases described as showing “intraepithelial lymphocytosis” that might be coeliac disease. **Methods**: To achieve this, we audited 18 months’ worth of duodenal biopsy reports in our centre. **Results**: A total of 6245 duodenal biopsies were identified, of which 73.76% were normal and at least 8.84% fell within the spectrum of coeliac disease. Additionally, 6.47% were classified as showing non-specific inflammation, 1.86% were adenomas, 0.45% were carcinomas, 0.06% were neuroendocrine tumours, 0.10% were lymphomas, and 0.03% were cases of flat dysplasia, giving a total of 0.64% of dysplastic or malignant diagnoses. Rarer diagnoses included ulceration, *Helicobacter pylori* infection, giardiasis, lymphangiectasia, transplant rejection, and lymphoma. Furthermore, 227 biopsies (3.63%) showed isolated intraepithelial lymphocytosis, of which 33 cases (14.5%) gave an overall clinicopathological picture of coeliac disease. **Conclusions**: We present the first long-term audit of all endoscopic duodenal biopsies received by the histopathology department of a tertiary-care facility. The results indicate that a fully automated diagnostic histopathology reporting system able to identify normal duodenal biopsies and biopsies within the spectrum of coeliac disease-associated enteropathy could decrease pathologists’ endoscopic duodenal biopsy workload by up to 80%.

## 1. Introduction

### 1.1. Biopsy Reporting Processes in Histopathology

Histopathology (often abbreviated to “pathology”) is a specialty that involves pathologists making diagnoses by looking at glass slides, prepared from patient biopsies, under the microscope. An electronic or paper clinical request form, with some, often rather limited, clinical details about the patient is received with the biopsy. The pathologist writes a microscopic description (i.e., what they see on the microscope slide, particularly focusing on the presence or absence of changes characteristic of particular diseases). They then provide a final diagnosis. Occasionally, the final diagnosis given is not that of a specific condition, but is a description of what is in the tissue, e.g., “intraepithelial lymphocytosis”, which means that there is an increase in the number of a type of white blood cell called a lymphocyte, in the innermost layer (the epithelium) of the intestine. While this change is often associated with coeliac disease [1,2,3], this appearance can be associated with conditions other than coeliac disease [4,5,6]. Receiving a biopsy description rather than a biopsy diagnosis can make it difficult for gastroenterologists, general practitioners, and other clinicians who treat patients to make a management decision.

### 1.2. Global Shortage of Pathologists

There exists a well-documented global shortage of pathologists resulting from a demand–supply mismatch [7,8,9,10,11]. A worldwide audit encompassing 162 countries averaged the number of pathologists per million of the population at 14 [12]. Only 3% of histopathology departments in the United Kingdom had sufficient staff to meet clinical demand in 2018 [13] and, in 2021, 10.8% of histopathology consultant posts were vacant [14]. Developing countries, in sub-Saharan Africa for example, have a disproportionately low number of trained pathologists dealing with a growing burden of cancer diagnosis [15], while resource-rich countries, like the United States of America, are seeing a steadily declining pool of pathologists in active clinical practice [10].

### 1.3. Effect on Endoscopic Duodenal Biopsies

With the existing workforce, endoscopic duodenal biopsies are frequently subject to delays as they are a relatively “low risk” sample type. As demonstrated in this study, only a minor proportion of the biopsies contain cancer. Backlogs in biopsy diagnosis can adversely impact patient care by prolonging decisions on management, delaying treatment initiation, and potentially extending hospital stays.

### 1.4. Diagnosing Coeliac Disease in Endoscopic Duodenal Biopsies

The most frequent specific diagnosis made on duodenal biopsies is gluten sensitivity or coeliac disease. Coeliac disease is an autoimmune disorder characterised by an adverse reaction to gluten, a protein found in wheat, barley, and rye, leading to small intestinal mucosal inflammation and malabsorption. The population prevalence of coeliac disease is estimated to be around 1%, yet only about one-third of these cases have been officially diagnosed [16,17]. Accurate diagnosis is particularly important for gluten-sensitive patients, as false-negative or equivocal results may leave them with chronic debilitating symptoms, including abdominal pain, diarrhoea, vomiting, malaise, fatigue, mouth ulcers, and itchy skin rashes, as well as long-term complications, including cancer, lymphoma, vitamin deficiency, anaemia, osteoporosis, and infertility [18,19].

### 1.5. Pathologists’ Discordance in Biopsy-Based Coeliac Disease Diagnosis

While a 3–6% discrepancy in agreement among histopathologists exists across all biopsy specimens [4,5,6], this discrepancy is notably higher for coeliac disease biopsies, as demonstrated by a recent study involving 17 histopathologists from the UK and Norway [20]. Disagreement between any two pathologists regarding the categorisation of a biopsy as histologically normal, containing coeliac disease, or showing so-called indeterminate enteropathy (inflammation without pathological changes specific enough to suggest a specific diagnosis) occurred 20% of the time (Cohen’s kappa of 0.59, considered “fair agreement”) [20], aligning with findings from other published studies [21,22,23,24,25,26].

Isolated intraepithelial lymphocytosis refers to an increase in a particular type of white blood cell in the inner layer of the duodenum, an abnormality that is most commonly seen in coeliac disease, but is not completely specific to that condition [18,27,28]. This further complicates the histology-based diagnosis of coeliac disease. While a lymphocyte count of more than 25 per 100 epithelial cells [1] is commonly recognised as indicative of coeliac disease [2,3], it lacks specificity in isolation without an appropriate clinical context, as similar elevations occur in so-called non-coeliac gluten intolerance, wheat and milk allergies, autoimmune diseases, or, uncommonly, as a reaction to certain medications [18,27,28]. The persistence of inaccurate and inconclusive diagnoses contributes significantly to the average period from symptom onset to a diagnosis of coeliac disease being 13.2 years in the UK [16,29]. This underscores the need for more rapid and accurate diagnostic approaches for duodenal biopsies.

In clinical practice, in cases of doubt about whether the histopathological diagnosis is coeliac disease, both the clinical picture and other laboratory parameters are considered [16,17]. Laboratory parameters include the coeliac disease-associated antibodies, immunoglobulin A (IgA) anti-tissue transglutaminase (tTG), and immunoglobulin G (IgG) anti-tTG and anti-endomysial antibody (EMA) [16,17]. In addition, the HLA genes, which play key roles in immune function, are highly polymorphic (genetically variable) between individuals. HLA genotypes suggesting a risk for coeliac disease have been identified, namely HLA-DQ2.2, HLA-DQ2.5, and HLA-DQ8 [16,17]. HLA typing is sometimes performed as part of the investigation of suspected coeliac disease.

### 1.6. The UK Biobank

The UK Biobank is a long-term prospective cohort study that includes genetic data associated with extensive phenotypic and health-related information. In this study we were able to use it to look at the UK population frequencies of coeliac disease-associated HLA alleles, as a comparator for those cases showing isolated intraepithelial lymphocytosis, which had undergone HLA typing [30,31,32,33].

### 1.7. Digital Pathology

With the growing utilisation of artificial intelligence in medical disciplines such as oncology, radiology, and dermatology [34,35,36], and its increasing application within histopathology [37,38], studies are under way to evaluate the practical implementation of computational diagnostic classifiers within routine clinical practice [39]. In many histopathology departments, glass microscope slides are now scanned digitally, providing an excellent opportunity to develop software to assist with or fully automate histopathological diagnosis. For duodenal biopsies these diagnoses are currently made by a pathologist looking at biopsy morphology, rather than by the use of any particular quantifiable biomarker.

### 1.8. Rationale for Audit

Our team is currently developing an artificial intelligence system for automating specific aspects of the histopathological evaluation of duodenal biopsies to improve diagnostic accuracy and workflow [40,41,42]. We are developing a “pathologist’s assistant” software tool that advises a pathologist of the probable diagnosis, likely halving time taken to diagnose a biopsy. Ultimately, we envisage this evolving into a fully automated tool. The data presented here show that if such a tool recognised only two diagnoses (coeliac disease and normal), it could decrease a pathologist’s workload by up to 80%. When developing a new diagnostic approach, it is crucial to understand what the expected results might look like following real-life deployment. This dataset is, therefore, exceptionally useful in estimating the percentages of the particular diagnoses one expects from duodenal biopsies. Significant deviation from these percentages could be used as a mechanism for monitoring the performance of any software tool making diagnoses on duodenal biopsy specimens.

We have been unable to find any studies auditing the spectrum and frequencies of diagnoses made on a large cohort of duodenal biopsies in a large UK hospital. Neither have we found any studies investigating the likelihood of an actual diagnosis of coeliac disease in biopsies designated “isolated intraepithelial lymphocytosis”. Furthermore, we have not found studies looking systematically at the spectrum of other diseases association with such biopsies.

Accordingly, we audited 18 months’ worth of duodenal biopsies received at Cambridge University Hospitals NHS Foundation Trust to determine the exact proportions of different diagnoses. Furthermore, pathologists’ diagnoses are frequently used as training data for AI models. Consequently, we analysed cases initially classified as showing isolated intraepithelial lymphocytosis, to determine how many eventually receive a coeliac disease diagnosis. If an AI dataset were built using these cases, they should be reclassified as coeliac disease, despite the initial pathologist’s report.

## 2. Materials and Methods

### 2.1. Duodenal Biopsy Case Identification

A retrospective review of the histological diagnoses of all endoscopic duodenal biopsies received by the Department of Histopathology, Addenbrooke’s Hospital, Cambridge, UK, was undertaken over a period of 18 months from 1 January 2018 to 30 June 2019 (full study methodology summarized in Figure 1). Inclusion criteria were endoscopic duodenal biopsies that were diagnosed during the 18-month time period, and, to ensure inclusivity and avoid any bias, there were no exclusion criteria. The full texts of the anonymised histopathology reports were retrieved for each patient. In order to take a snapshot of current histopathology practice, no re-review of microscope slides was conducted and no interobserver variability needed to be considered. As this was an anonymised audit project, it is not possible to make the dataset publicly available. While the data herein will guide the development of an AI system, no AI was used in the methods of this study.

### 2.2. Analysis of the Duodenal Biopsy Cases Identified

For each case, the text of the reporting pathologist’s final diagnosis was used to determine the diagnostic category for each biopsy, and the percentages of each category were calculated (Table 1, Figure 2).

### 2.3. Identification of Key Words Indicative of Serious Pathology

For rare or more serious conditions that may not be safely assessed in an automated way in the near future (e.g., refractory coeliac disease, dysplasia, malignancy, and ulceration), the clinical details accompanying the specimen were examined in order to determine key words that could be used to exclude such cases from a potential AI-based automated diagnostic tool (Table 2).

### 2.4. Demographic and Clinical Features Associated with Isolated Intraepithelial Lymphocytosis

All cases with a histological description of isolated intraepithelial lymphocytosis were subjected to a clinical chart review. We recorded demographic features (age and sex) and the clinical indication(s) for biopsy received on the request form (Table 3).

### 2.5. Attempted Refinement of Diagnosis in Cases of Isolated Intraepithelial Lymphocytosis

For biopsies with isolated intraepithelial lymphocytosis, we determined how many patients in this cohort were subsequently clinically diagnosed with coeliac disease by using blood results (coeliac disease-specific autoantibody levels (anti-tTG (IgA or IgG) and EMA)) and clinical follow-up data (Table 4). Where individuals had had a genetic analysis of their HLA type (a genetic locus associated with many immunological conditions), we compared the at-risk HLA genotypes for coeliac disease in this audit subgroup with those of the HLA genotype population frequency, as determined in the genome-wide genotyping dataset from the UK Biobank’s anonymized database available on their Research Analysis Platform (RAP) [30,31,32,33].

### 2.6. Calculated of Expected UK Population Frequency of Coeliac-Risk HLA Types Using the UK Biobank

We obtained HLA typing data from 32,856 patients from the UK Biobank [30,31]. We split the patients into a coeliac disease group (*n* = 3094) and a control group (*n* = 29.762) based on their responses to the online coeliac disease and dietary questionnaires, as well as their health records. To identify genetic risk factors, we created an HLA typing code by using the UK Biobank’s imputation threshold [32]. This code leveraged specific allele combinations to identify HLA genotypes associated with an increased risk of coeliac disease [33]. We thus identified all patients with one or more of the following coeliac disease-risk alleles: HLA-DQ2.2, HLA-DQ2.5, and HLA-DQ8 (Appendix A).

### 2.7. Determination of Conditions Associated with Isolated Intraepithelial Lymphocytosis

We further reviewed our cohort of patients’ available medical records to determine whether a patient had any reported systemic, inflammatory, immunological, autoimmune, gastrointestinal, or hepatobiliary condition (Table 5).

### 2.8. Literature Review of Conditions Associated with Isolated Intraepithelial Lymphocytosis

We searched the PubMed database (https://pubmed.ncbi.nlm.nih.gov/ accessed on 15 June 2023) for each of the conditions associated with the histological finding of isolated intraepithelial lymphocytosis in our patient cohort, in order to obtain an approximate UK population frequency of these (Table 5).

### 2.9. Recording of Other Abnormal Laboratory Investigations in Patients with Isolated Intraepithelial Lymphocytosis

For biopsies with isolated intraepithelial lymphocytosis, we determined how many had had an abnormal laboratory result for any parameter related to inflammation, the immune system, any autoimmune condition, or any gastrointestinal condition (Table 6).

## 3. Results

The findings of this audit are presented in three parts: (1) the diagnostic categories and frequency of biopsies in each category; (2) the key words identifying the minor diagnostic categories unlikely, for patient safety reasons, to be amenable to automated diagnosis; and (3) an exploration of the characteristics of cases showing isolated intraepithelial lymphocytosis.

### 3.1. Summary of Biopsy Diagnostic Categories

In total, 6245 endoscopic duodenal biopsies were received over the 18-month period (Table 1). The proportions of the distinct diagnoses, derived from the histopathology reports, are illustrated in Figure 2.

The two largest diagnostic categories were those of normal histology, with 4606 (73.76%) biopsies, and active coeliac disease, with 344 (5.50%) biopsies. Additionally, within the spectrum of coeliac disease-related pathology, eight (0.13%) were diagnosed as villous atrophy alone, five (0.08%) as villous atrophy with raised intraepithelial lymphocytes, and three (0.05%) as gluten-sensitive enteropathy. A total of 172 (2.75%) biopsies were from patients previously diagnosed with coeliac disease who were on a gluten-free diet (GFD). Of these, 119 (1.90% of the total; 69.2% of the 172 on a gluten-free diet) biopsies displayed complete healing of duodenal mucosa, while 53 (0.85% of total; 30.8% of the 172 on a gluten-free diet) showed only partial mucosal recovery. Finally, 20 (0.32%) biopsies were diagnosed as refractory coeliac disease. Hence, a total of 552 biopsies (8.84%) exhibited features associated with coeliac disease, some probably representing similar conditions given semantically different labels, spanning a histomorphological spectrum from active disease status to normal histology.

A total of 227 (3.63%) biopsies were reported as having isolated intraepithelial lymphocytosis (commonly considered as more than 25 lymphocytes per 100 epithelial cells) [1], with a comment advising the treating physician to correlate biopsy findings with clinical information to determine the diagnosis. The diagnosis of “non-specific inflammation” was given to 404 (6.46%) biopsies. This category included patients presenting with a diverse array of reported symptoms, with the biopsy showing mixed chronic or active chronic inflammation in the mucosa and/or lamina propria. Other notable groups were adenomas, comprising 116 (1.86%) of the cases, focal gastric heterotropia, with 104 (1.66%) cases, and gastric metaplasia, with 48 (0.77%) cases.

In total, 28 biopsies (0.45%) were diagnosed with carcinoma, which included 21 cases of primary duodenal adenocarcinomas and 7 cases of metastatic carcinoma. Two biopsies (0.03%) showed flat (non-adenomatous) dysplasia, which is a precursor to carcinoma. There were four (0.06%) biopsies of neuroendocrine tumour and six biopsies (0.10%) showing lymphoma. Thus, in total, 40 cases (0.64% of total duodenal biopsies) in an 18-month period demonstrated malignancy or pre-malignancy.

Minor diagnostic categories included lymphangiectasia, with 25 (0.40%) cases, and ulceration, with 25 (0.40%) cases. There were 20 (0.32%) biopsies from intestinal transplant recipients showing healthy graft status (15 biopsies, 0.24%), mild rejection (two biopsies, 0.032%), bacterial infection (two biopsies, 0.032%), and post-transplant lymphoproliferative disorder (one biopsy, 0.016%). Additionally, seven (0.11%) duodenal biopsies were from peripheral blood, bone marrow, or renal transplant recipients demonstrating graft-versus-host disease. Furthermore, there were twelve (0.19%) biopsies diagnosed with giardiasis, ten (0.16%) cases of mucosal erosion, nine (0.14%) cases showing *Helicobacter pylori* infection, seven (0.11%) cases each of regenerative mucosal changes and hyperplasia without dysplasia, six (0.10%) cases of lipoma, five (0.08%) cases of drug-induced enteropathy, five (0.08%) cases of Crohn’s disease, four (0.06%) cases of viral enteropathy, three (0.05%) cases with lymphoid aggregates, and two (0.03%) cases with granulomas. Finally, the following diagnoses contributed one (0.02%) case each: common variable immunodeficiency syndrome, amyloidosis, thrombus, pseudolipomatosis, spindle cell tumour, pseudomelanosis, and gangliocytic paraganglioma.

### 3.2. Key Words for Minor Diagnostic Categories

Each biopsy request is generally accompanied by clinical information/the indication(s) for biopsy from the treating physician. Commonly mentioned clinical features of coeliac disease include iron deficiency anaemia, weight loss, diarrhoea, abdominal pain, dyspepsia, and/or nutrient deficiency. Exploring this aspect, we identified certain key words used in the clinical request forms accompanying the biopsy that suggested abnormal conditions other than coeliac disease. If utilising a potential automated diagnostic tool, these key words could act as indicators for triaging biopsies and prioritising them for evaluation by a pathologist (Table 2).

### 3.3. Isolated Intraepithelial Lymphocytosis

Intraepithelial lymphocytosis is a descriptive histological feature rather than a distinct diagnostic entity. It is most commonly associated with coeliac disease, although it can be seen in a range of other conditions [18,27,28]. We sought to determine the percentage of cases of intraepithelial lymphocytosis likely to be coeliac disease, by considering what other evidence of coeliac disease was available in a patient’s record. We also considered other possible associated conditions, particularly systemic, inflammatory, immunological, autoimmune, gastrointestinal, and hepatobiliary conditions, by looking for both clinical evidence and laboratory test data to support these diagnoses. Accordingly, the group of 227 biopsies with isolated intraepithelial lymphocytosis was subjected to a chart review. Iron deficiency anaemia was found to be the most common clinical presentation, along with a variety of gastrointestinal symptoms, weight loss, and vitamin deficiency (B12 and folate) (Table 3). In 23 out of 227 (10.13%) cases, no clinical information was received with the biopsy. The clinical presentations appeared unlikely to be sufficiently specific to suggest a particular diagnosis.

To assess the potential presence of (undiagnosed) coeliac disease, we analysed these 227 cases of isolated intraepithelial lymphocytosis (Table 4). Among these cases, 24 patients eventually received a coeliac disease diagnosis either before or after the biopsy. Specifically, 9 cases (3.96%) had already been diagnosed with coeliac disease prior to the biopsy and were already following a gluten-free diet, 13 cases (5.73%) were diagnosed via subsequent clinicopathological correlation, and 2 cases (0.88%) received a coeliac disease diagnosis following an additional biopsy. Moreover, nine biopsies (3.96%) showed coeliac disease-related laboratory results: five (2.20%) exhibited clear positive findings for both tissue transglutaminase IgA (tTG) and endomysial antibody (EMA), while four cases (1.76%) had a positive IgA tTG result with a negative or missing EMA result (Table 4). While patients are tested for IgG tTG, if they have a negative test for IgA tTG and are suspected of being IgA-deficient, none of the patients in our isolated intraepithelial lymphocytosis cohort had been tested for IgG tTG. These results would indicate that, at minimum, 24 (10.6%) of the 227 patients with isolated intraepithelial lymphocytosis are likely to have coeliac disease.

In our data from the UK Biobank, 98% of patients with coeliac disease carried at least one of the HLA-DQ2.2, HLA-DQ2.5, or HLA-DQ8 coeliac disease-risk alleles (Appendix A), consistent with other studies [16,17]. HLA typing data were available for 21 of these patients, of whom 14 (67%) carried at least one coeliac disease-associated HLA type (Table 4), compared with the general population frequency of 58% that we calculated from the UK Biobank (Appendix A). Of these 21, 4 were diagnosed with coeliac disease following a biopsy based on alternative criteria, and all 4 carried a coeliac disease-associated HLA type (Table 4). Among the remaining 17 patients, who had no other evidence supporting a coeliac disease diagnosis, 10 (59%) carried a coeliac disease-associated HLA type (Table 4). This closely mirrors the prevalence of coeliac disease-associated HLA types in the non-coeliac disease control group from the UK Biobank (58%), meaning that it cannot be used to comment on the probability of coeliac disease in this group of patients (see Appendix A). These findings underscore the need for thorough investigation of raised IEL cases to ensure that all patients who eventually receive a coeliac disease diagnosis are correctly classified when developing high-quality AI training datasets.

We further examined the patient record for reported systemic, inflammatory, immunological, autoimmune, gastrointestinal, and hepatobiliary conditions in our isolated intraepithelial lymphocytosis patient cohort. These patients had an increased prevalence of inflammatory/autoimmune conditions and their sequelae, including hypothyroidism, Graves’ disease, microscopic colitis, pernicious anaemia, Crohn’s disease, ankylosing spondylosis, ulcerative colitis, and Addison’s disease, compared to the general population (Table 5). We estimated the frequency of these conditions in the general population by means of a PubMed literature review. A significant portion of these cases of isolated intraepithelial lymphocytosis remained without a final diagnosis despite a range of abnormal laboratory tests, elevated faecal calprotectin being the most frequent finding (Table 6). Finally, for 119 of the 227 cases (52.42%) no other clinical or laboratory signifier was available, presenting a definitive diagnostic challenge.

## 4. Discussion

This is the first large, comprehensive audit study of the spectrum and frequencies of diagnoses made in a sizeable cohort of duodenal biopsies (*n* = 6245) in a large UK hospital. It is also the first study systematically investigating the likelihood of a diagnosis of coeliac disease, or other associated specific conditions, in biopsies designated “isolated intraepithelial lymphocytosis”.

We showed here that the majority of endoscopic duodenal biopsies are either normal (73.76%) or show coeliac disease-associated pathology (at least 8.84%, as discussed below), meaning that these two diagnoses alone constitute at least 82.60% of all duodenal biopsies. Automating the diagnosis of just these two categories could potentially offload > 80% of pathologists’ duodenal biopsy workloads, providing a realistic means of mitigating the shortage of pathologists and preventing backlogs associated with endoscopic duodenal biopsies.

When considering a novel diagnostic strategy, prioritising safety is paramount. The fact that only 0.64% of duodenal biopsies contain dysplasia (pre-cancer), carcinoma, neuroendocrine tumour, or lymphoma (cancer) makes this biopsy type one of the safest for instigating fully automated diagnosis. An AI-based image analysis approach that solely identifies the two most frequent diagnostic categories (coeliac disease and normal) could further enhance the clinical safety of such a system. It is also more achievable than an approach that attempts to recognise all major and minor pathologies. Here we have shown that only small numbers of biopsies showing other diagnoses are encountered, meaning that there are insufficient numbers of examples of these with which to train and test an AI system. One might thus envisage a system that recognises the features of normal and coeliac disease duodenal biopsies, and triages all remaining biopsies to a pathologist.

Although there are only very small numbers of duodenal biopsies containing evidence of very serious conditions, including ulceration, pre-cancer, and cancer (Table 1), misdiagnosing such a condition as either normal or coeliac disease could significantly harm a patient. Therefore, as well as the image analysis system triaging abnormal non-coeliac biopsies to a pathologist, it would be preferable to have the additional safety feature of a pre-analysis system to flag biopsies likely to contain such conditions. Having multiple layers of redundancy built into a system in this way would enhance clinical safety. The initial flagging of biopsies might be achieved by the automated scrutiny of the words in the clinical details received with the biopsy.

Our examination of the clinical details associated with abnormal, non-coeliac biopsies allowed us to compile key words related to potentially rapidly harmful conditions. These terms could be used to triage biopsies for immediate evaluation by a pathologist. The presence of polyps or tumours, gastrointestinal bleeding, jaundice, biliary obstruction, underlying systemic diseases, immunosuppression, hereditary cancer syndromes, and prior transplants indicated diagnoses with poorer prognosis. Therefore, any biopsy received with clinical information containing these key words should be flagged for immediate examination by a pathologist. Notably, the key words for the parasite infection giardiasis overlapped with those commonly used for coeliac disease. To automate the diagnosis of suspected coeliac disease biopsies, a recognition mechanism for Giardia parasites in histological sections might need to be developed in parallel. One major caveat is that this key-word-based triage method is reliant on the provision of comprehensive clinical details, which does not happen for all biopsies, presenting a challenge to its practical implementation.

In our review of 227 biopsies diagnosed as isolated intraepithelial lymphocytosis, which is a descriptive histological finding rather than a distinct diagnostic entity, we sought to identify potential underlying coeliac disease and related conditions. Iron deficiency anaemia emerged as the most common clinical presentation. However, iron deficiency, various gastrointestinal symptoms, weight loss, and vitamin deficiencies seen in the isolated intraepithelial lymphocytosis patient cohort would not be specific enough to suggest coeliac disease or any other specific diagnosis.

In total, 24 (10.6%) of the 227 patients with isolated intraepithelial lymphocytosis eventually received a clinical diagnosis of coeliac disease, either before or after the biopsy. Coeliac disease-related laboratory abnormalities were identified in an additional 9 cases, with positive tissue transglutaminase IgA (tTG) or endomysial antibody (EMA) results, meaning that, in total, at least 33 (14.5%) of the 227 patients with isolated intraepithelial lymphocytosis actually had coeliac disease. Interestingly, patients with isolated intraepithelial lymphocytosis also exhibited a higher prevalence than the general population of autoimmune and inflammatory disorders that can be associated with coeliac disease [18,19], e.g., thyroid disease, Graves’ disease, rheumatoid arthritis, pernicious anaemia, microscopic colitis, and Addison’s disease (Table 5). Corroborating these associations, a number of patients in this cohort had positive laboratory tests for autoimmune and inflammatory conditions (Table 6).

While the largest single diagnostic category among patients with isolated intraepithelial lymphocytosis appears to be coeliac disease at 14.5%, the next most common two associations were with two commonly used medications. Proton pump inhibitors were likely responsible for 26 out of 227 cases (11.45%), and non-steroidal anti-inflammatory drugs for 18 out of 227 cases (7.93%). This means that, overall, medications may be responsible for 19.4% of cases with isolated intraepithelial lymphocytosis. Taken together with the 5 (0.08%) cases that were given a confident diagnosis of drug-induced enteropathy, 0.78% of total duodenal biopsies (49 out of 6245) showed changes that were likely to be drug (medication)-related. This contrasts with the much higher likely total percentage of cases of coeliac disease-related pathology (585 out of 6245 cases, 9.37%), made up of the 552 (8.84%) given a confident coeliac disease spectrum diagnosis initially and the 33 (0.53%) additional biopsies identified within the isolated intraepithelial lymphocytosis cohort.

When biopsies were diagnosed by a pathologist, 3.63% of biopsies (227 out of 6245) were placed in the isolated intraepithelial lymphocytosis category, rather than being given a specific diagnosis. It remains to be seen whether or not a carefully trained, AI-based diagnostic tool will be able to separate cases of isolated intraepithelial lymphocytosis into diagnostic subgroups, for example, identifying features not readily appreciated by a human observer that can distinguish coeliac disease from other causes.

Despite our detailed analysis, a significant proportion of cases remained without a definitive diagnosis. Overall, these results underscore the diagnostic challenges associated with isolated intraepithelial lymphocytosis, highlighting the need for comprehensive investigations to ensure accurate classification, particularly when developing high-quality AI training datasets.

One limitation of this study is the fact that the diagnostic proportions may not be fully generalisable to other hospitals or other countries, due to variations in case mix and protocols leading to patient referral for endoscopy, as well as age, ethnicity, and socioeconomic factors of the patient population served by other hospitals. However, this study shows the feasibility of performing a similar audit to determine the proportion of various diagnoses in other clinical settings.

## 5. Conclusions and Future Directions

We have applied a simple audit approach to a large cohort of duodenal biopsies, and our results show that a system capable of identifying most biopsies with normal mucosa and those with features of coeliac disease might achieve automated diagnosis of around 80% of duodenal biopsies. These data can be utilised for designing a strategy for the AI-mediated automation of duodenal biopsy diagnosis. Such an approach would save substantial pathologist time, mitigating the effects of pathologist shortages and increased workload complexity as well as volume. This could improve biopsy turnaround times, not just for duodenal biopsies but for other specimen types, as it would free up pathologists’ time to tackle these. The disadvantages of developing such an AI-mediated automatic reporting system are the expense, time, expertise, and regulatory approval required to do so, as well as the risk of error, leading to discussions about whether the developer of such a system or the healthcare organisation running it would be responsible for such an error. Notwithstanding, our data provide a strong case for automating the diagnosis of duodenal biopsy histopathology. Our results additionally provide a baseline for the assessment of the expected output of such an AI-mediated system, facilitating the ongoing audit of the diagnostic outputs of any such system following clinical implementation. Due to differences in case mix and endoscopy referral practices, it may be necessary to undertake a small baseline audit in each centre and/or country before introducing automation to the process of duodenal biopsy diagnosis. This study demonstrates the feasibility and utility of undertaking an audit of this nature.

## Figures and Tables

**Figure 1 diagnostics-15-01483-f001:**
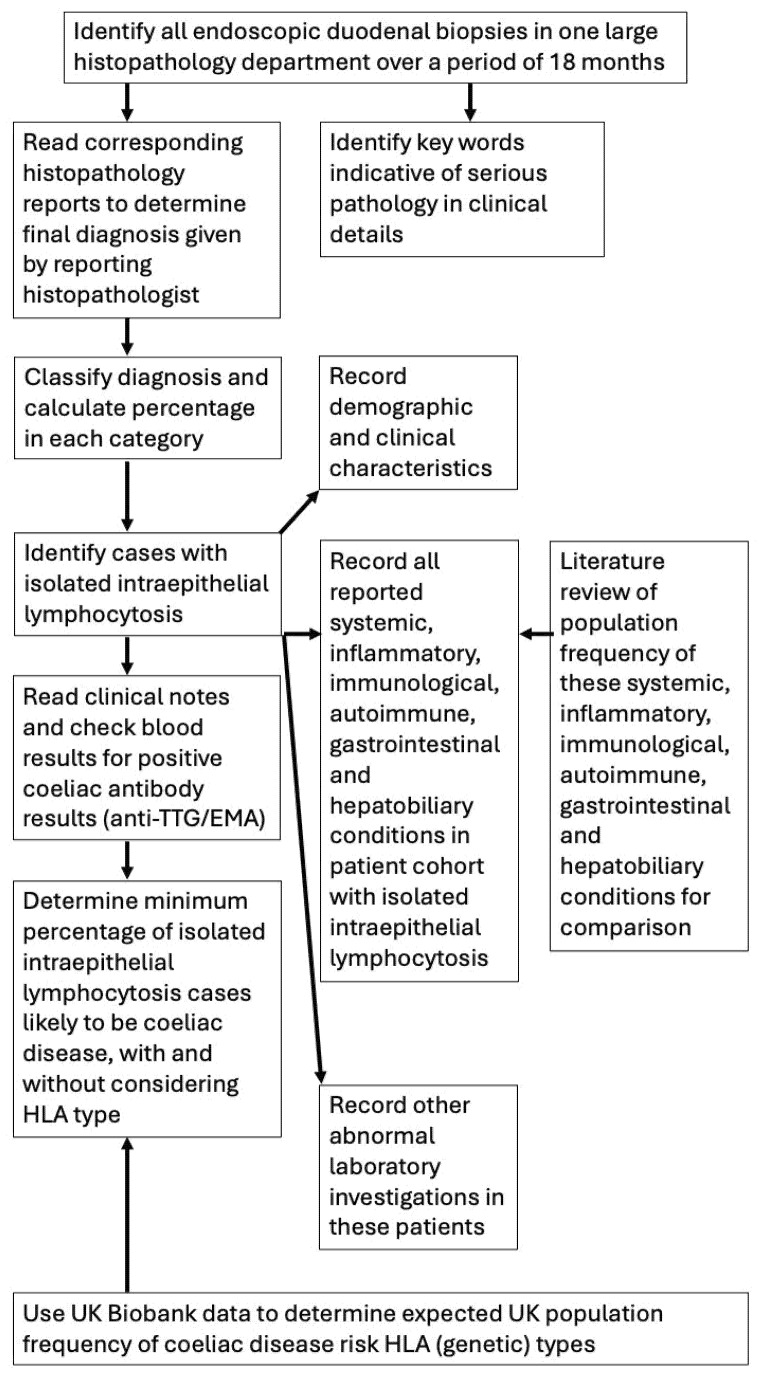
Summary of study methodology.

**Figure 2 diagnostics-15-01483-f002:**
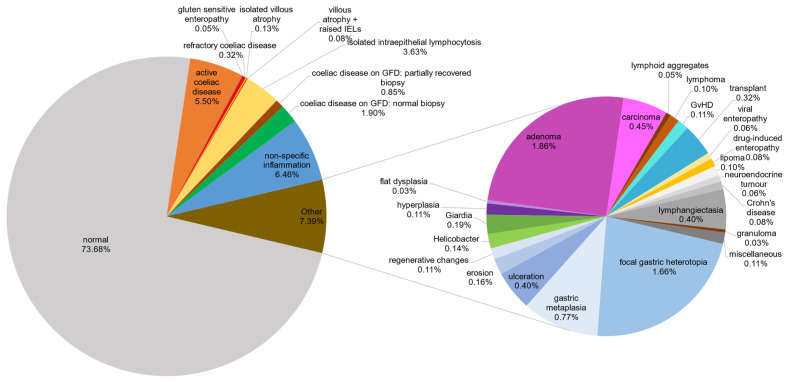
Pie charts depicting major and minor diagnostic categories. This pie-within-pie chart visualises the distribution of the distinct pathological categories outlined in Table 1. The distribution of normal biopsies, coeliac-associated biopsy categories, and non-specific inflammation is represented in the larger pie chart, with a detailed breakdown of abnormal, non-coeliac diagnostic categories depicted in the smaller embedded pie chart. The segment labelled “miscellaneous” in the smaller pie chart comprises minor diagnostic categories containing one case each (common variable immunodeficiency syndrome, amyloidosis, thrombus, pseudolipomatosis, spindle cell tumour, pseudomelanosis, and gangliocytic paraganglioma). These diagnoses were made by the pathologists examining biopsy morphology, rather than by using particular biomarkers. Re-examination of the biopsies was not undertaken. For a definition of the terminology in this figure, please refer to Appendix A.

**Table 1 diagnostics-15-01483-t001:** Distribution of diagnostic categories. The total frequency (*n*, %) of biopsies belonging to each distinct diagnostic category over the audit period of 18 months is shown. The diagnostic category was determined using the original pathologist’s diagnosis in the histopathology report corresponding to each biopsy case. For a definition of the terminology in this table, please refer to Appendix A.

Classification	Total (*n*)	Percentage (%)
Normal	4606	73.76
Coeliac-associated diagnoses	552	8.84
Active coeliac disease	344	5.50
Coeliac disease on GFD ^1^: normal biopsy	119	1.90
Coeliac disease on GFD: partially recovered biopsy	53	0.85
Refractory coeliac disease	20	0.32
Isolated villous atrophy	8	0.13
Villous atrophy + raised IELs ^2^	5	0.08
Gluten-sensitive enteropathy	3	0.05
Isolated intraepithelial lymphocytosis	227	3.63
Non-specific inflammation	404	6.46
Neoplastic changes		
Adenoma	116	1.86
Carcinoma	28	0.45
Lymphoma	6	0.10
Lipoma	6	0.10
Neuroendocrine tumour	4	0.06
Flat dysplasia	2	0.03
Infections		
Giardiasis	12	0.19
*Helicobacter pylori* infection	9	0.14
Viral enteropathy	4	0.06
Benign gastric epithelium-related changes		
Focal gastric heterotopia	104	1.66
Gastric metaplasia	48	0.77
Duodenal mucosal surface changes		
Ulceration	25	0.40
Lymphangiectasia	25	0.40
Erosion	10	0.16
Regenerative changes	7	0.11
Hyperplasia	7	0.11
Autoimmune/Inflammatory conditions		
Crohn’s disease	5	0.08
Drug-induced enteropathy	5	0.08
Lymphoid aggregates	3	0.05
Granuloma	2	0.03
Transplant-related diagnoses		
Transplant	20	0.32
Graft versus host disease (GvHD)	7	0.11
Miscellaneous	7	0.11
**Total**	**6245**	**100**

^1^ GFD—gluten-free diet. ^2^ IEL—intraepithelial lymphocytes.

**Table 2 diagnostics-15-01483-t002:** Key words associated with certain abnormal, non-coeliac diagnostic entities. For a definition of the terminology in this table, please refer to Appendix A.

Biopsy Diagnostic Category *	Clinical Key Words
Adenoma	Gastrointestinal bleeding, familial adenomatous polyposis (FAP), and polyp or papillary tumours on endoscopy
Amyloidosis	Systemic infiltrative disease
Carcinoma	Jaundice, gastrointestinal bleeding, bowel obstruction, mass or dilated bile duct on imaging or endoscopic examination, and previous diagnosis of malignancy of any organ
Common variable immunodeficiency	Background of immunodeficiency syndrome, opportunistic infections (norovirus in our data)
Crohn’s disease	Previous diagnosis of Crohn’s disease, stricture, and fistula
Drug-induced enteropathy	Established autoimmune disease on immunomodulators/NSAIDs ^†^
Flat (non-adenoma) dysplasia	Gastrointestinal bleeding, necrotic ulcer
Focal gastric heterotopia	Polyp on endoscopy
Gastric metaplasia	Polyp on endoscopy
Giardia	Iron deficiency anaemia, weight loss, and abdominal pain
Granuloma	History of tuberculosis or Crohn’s disease
Hyperplasia without dysplasia	Polyp on endoscopy
Lipoma	Nodule on endoscopy
Lymphangiectasia	Gastrointestinal bleeding, ill-defined mass on endoscopy
Lymphoid aggregates	Mass on endoscopy
Lymphoma	Duodenal mass, gastrointestinal bleeding, and previous diagnosis of lymphoma
Neuroendocrine tumour	Nodule/polyp
Ulceration	Gastrointestinal bleeding, abnormal barium study, and relevant drug history (NSAIDs ^†^)
Viral enteropathy	Immune suppression

* This list is not exhaustive due to some diagnostic categories having too few numbers and/or inadequate clinical information. ^†^ NSAID—non-steroidal anti-inflammatory drug.

**Table 3 diagnostics-15-01483-t003:** Demographic characteristics and indications for biopsy in patients with isolated intraepithelial lymphocytosis.

Characteristic	Finding (*n* = 227)
Median age in years (standard deviation)	49 (17.7)
Female sex (%)	163 (71.8)
Male sex (%)	64 (28.2)
Indications for Biopsy (%)	
Iron deficiency anaemia	108 (47.6)
Diarrhoea	44 (19.4)
Abdominal pain	32 (14.1)
Weight loss	33 (14.5)
Dyspepsia	24 (10.6)
Reflux	16 (7.1)
Nausea or vomiting	9 (4.0)
Folate deficiency	9 (4.0)
Non-specific altered bowel movements	8 (3.5)
Dysphagia	8 (3.5)
Bloating	7 (3.1)
Constipation	4 (1.8)
Vitamin B12 deficiency	4 (1.8)
No information provided	23 (10.1)

**Table 4 diagnostics-15-01483-t004:** Features suspicious for coeliac disease in 33 (28.94%) of 227 cases of isolated intraepithelial lymphocytosis. The percentage stated in brackets is relative to the 33 cases with features suspicious of coeliac disease.

Characteristic	Finding (*n* = 33)
Median age in years (standard deviation)	48 (16.6)
Female sex (%)	23 (69.7)
Male sex (%)	10 (30.3)
Integrated clinical diagnosis of coeliac disease (%)	*n* = 24
Prior to biopsy	9 (27.3)
Subsequent to biopsy	13 (39.4)
After further biopsy	2 (6.1)
Features strongly suspicious for coeliac disease (%)	*n* = 9
Positive IgA(tTG) and positive EMA	5 (15.2)
Positive IgA(tTG) and negative/missing EMA	4 (12.1)

**Table 5 diagnostics-15-01483-t005:** Prevalence of co-morbidities among patients in our isolated intraepithelial lymphocytosis cohort, compared with reported UK population prevalence. For a definition of the terminology in this table, please refer to Appendix A.

Associated Condition	Number of Cases, *n* (%) (N = 227)	Prevalence in UK Population (From Published Studies)
Proton pump inhibitor use	26 (11.45)	18.04% [43]
NSAID ^1^ use	18 (7.93)	Variable [44]
*Helicobacter pylori* infection	12 (5.29)	Up to 35% [45]
Irritable bowel syndrome	9 (3.96)	10–20% [46]
Hypothyroidism	8 (3.52)	2% [47]
Graves’ disease	6 (2.64)	0.75% [48]
Microscopic colitis	4 (1.76)	0.1% [49]
Pernicious anaemia	3 (1.32)	0.05–0.2% [50]
Crohn’s disease	3 (1.32)	0.27% [51]
Rheumatoid arthritis	3 (1.32)	1% [52]
Ankylosing spondylitis	3 (1.32)	0.05–0.2% [53]
Gastrointestinal adenocarcinoma	3 (1.32)	Variable [54,55]
Ulcerative colitis	2 (0.88)	0.39% [51]
Seronegative spondyloarthropathy	2 (0.88)	0.5–1.9% [56]
Addison’s disease	2 (0.88)	0.01% [57]
Transplanted ileum and colon	1 (0.44)	19 cases in 2018–19 [58]
Duodenal neuroendocrine tumour	1 (0.44)	0.00017% [59]

^1^ NSAID—non-steroidal anti-inflammatory drug.

**Table 6 diagnostics-15-01483-t006:** Miscellaneous abnormal laboratory investigations in patients with isolated intraepithelial lymphocytosis.

Abnormal Laboratory Test	Likely Significance of Test	Number of Cases, *n (%)*
Elevated faecal calprotectin	Raised in inflammatory bowel disease (e.g., Crohn’s disease, ulcerative colitis), but can be raised in many inflammatory gastrointestinal conditions, including coeliac disease and infection	37 (16.3)
Elevated ESR ^1^	Raised in many inflammatory and autoimmune conditions, but also in infections and cancer	17 (7.5)
Elevated CRP ^2^	Raised in many inflammatory and autoimmune conditions, but also in infections and tissue injury	15 (6.6)
IgA ^3^ deficiency	A common immunodeficiency, which may be asymptomatic, or associated with recurrent respiratory infections, allergies, and autoimmune conditions	10 (4.4)
Elevated ANA ^4^	A relatively non-specific marker of autoimmune disease	8 (3.5)
Elevated rheumatoid factor	A relatively non-specific marker of autoimmune disease, particularly rheumatoid arthritis	7 (3.1)
Elevated p-ANCA ^5^	A marker of autoimmune disease, particularly, but not exclusively, those affecting blood vessels (vasculitis) and the kidney (glomerulonephritis)	3 (1.3)
Elevated anti-cyclic citrullinated protein	A highly specific marker for rheumatoid arthritis	2 (0.9)
Elevated c-ANCA ^6^	A marker of autoimmune disease, particularly, but not exclusively, those affecting blood vessels (vasculitis) and the kidney (glomerulonephritis)	1 (0.4)
Elevated anti-U1 RNP ^7^	A marker strongly associated with a group of autoimmune conditions, known as mixed connective tissue disease	1 (0.4)

^1^ ESR—erythrocyte sedimentation rate. ^2^ CRP—C-reactive protein. ^3^ IgA—immunoglobin A. ^4^ ANA—antinuclear antibody. ^5^ p-ANCA—perinuclear antineutrophil cytoplasmic antibody. ^6^ c-ANCA—cytoplasmic antineutrophil cytoplasmic antibody. ^7^ Anti-U1 RNP—antibody to U1 ribonucleoprotein.

## Data Availability

The original contributions presented in the study are included in the article; further inquiries can be directed to the corresponding author. This study includes data from the UK Biobank, a major biomedical database (https://www.ukbiobank.ac.uk/ accessed 1 July 2023).

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
