# Peer review of "Duodenal Biopsy Audit: Relative Frequency of Diagnoses, Key Words on Request Forms Indicating Severe Pathology, and Potential Diagnoses for Intraepithelial Lymphocytosis, as a Foundation for Developing Artificial Intelligence Diagnostic Approaches"

_diagnostics, 2025, doi:10.3390/diagnostics15121483_

Round 1
Reviewer 1 Report
Comments and Suggestions for Authors
This study includes a strong research and analysis process for the development of artificial intelligence-supported diagnostic systems. 6245 endoscopic biopsy data obtained over an eighteen-month period were reported and analyzed in detail. Although the article is interesting in terms of content, it is thought that it needs to be strengthened in some aspects. I suggest the following corrections for this study:
1-The main contribution of the study should be emphasized more clearly. In particular, the main purpose of the study and the obtained outputs can be presented more clearly in the introduction section of the article. In its current form, the introduction section summarizes the pathology processes and the general situation well; however, the original contribution of this study should be emphasized more strongly.
2-A visual summarizing the flow of the study can be added. A diagram reflecting the research and methodology process can enable the reader to follow the process more easily. It is also recommended to increase the readability of the tables. The flow of the article is quite straightforward; therefore, there is a risk of losing the reader's interest.
3-The issue of "applicability of the AI ​​system" should be addressed more clearly. It is seen that a strong data set has been created in the article. Brief and clear statements can be made about whether this dataset will be shared or not and the data sharing policy. In the discussion section, the potential advantages and limitations of the AI ​​system can also be briefly evaluated. If possible, it would be useful to add a sample AI application scenario.
4- Although it is important to preserve academic language, it would be useful to explain some technical terms. The article is addressed to researchers who are competent in the medical field, but it also has a structure that is open to multidisciplinary studies. Therefore, it is recommended to provide brief explanations of some medical terms, especially when they are first mentioned.
Comments on the Quality of English Language-
Author Response
GENERAL RESPONSE: We thank this reviewer for working carefully through our manuscript and providing constructive and helpful suggestions, all of which we have implemented, overall substantially improving the clarity and readability of our submission.
This study includes a strong research and analysis process for the development of artificial intelligence-supported diagnostic systems. 6245 endoscopic biopsy data obtained over an eighteen-month period were reported and analyzed in detail. Although the article is interesting in terms of content, it is thought that it needs to be strengthened in some aspects. I suggest the following corrections for this study:
1- The main contribution of the study should be emphasized more clearly. In particular, the main purpose of the study and the obtained outputs can be presented more clearly in the introduction section of the article. In its current form, the introduction section summarizes the pathology processes and the general situation well; however, the original contribution of this study should be emphasized more strongly. RESPONSE: The title has been amended and the introduction and discussion have been reworded to take account of this.
2-A visual summarizing the flow of the study can be added. A diagram reflecting the research and methodology process can enable the reader to follow the process more easily. It is also recommended to increase the readability of the tables. The flow of the article is quite straightforward; therefore, there is a risk of losing the reader's interest. RESPONSE: Adding a schematic is an excellent idea, which we have taken on board (and creating it helped us see which parts of the study were inadequately described in the methods section (now rectified)). We have added a new figure (figure 1) to summarise the flow of the study. While we appreciate that the tables contain a lot of medical terminology, their structures are extremely simple, and it is not possible to simplify them any further.
3- The issue of "applicability of the AI ​​system" should be addressed more clearly. It is seen that a strong data set has been created in the article. Brief and clear statements can be made about whether this dataset will be shared or not and the data sharing policy – all we have is the pathology reports, which will not be shared. In the discussion section, the potential advantages and limitations of the AI ​​system can also be briefly evaluated. If possible, it would be useful to add a sample AI application scenario. RESPONSE: We have made it clearer in the title, abstract and introduction and discussion that we are planning to apply an AI system to the problem of histopathological diagnosis of duodenal biopsies, and have made it clearer that the AI system would either function as a pathologist’s assistant tool or would undertake fully automated diagnostic histopathological reporting, by way of a sample AI application scenario. The work described here is merely to guide the expected percentages/ ratios of various diagnosis that we would expect to see as the output of such an AI system. We have added the following to the first paragraph of the methods section: “As this was an anonymised audit project, it is not possible to make the dataset publicly available. While the data herein will guide the development of an AI system, no AI was used in the methods of this study.”
4- Although it is important to preserve academic language, it would be useful to explain some technical terms. The article is addressed to researchers who are competent in the medical field, but it also has a structure that is open to multidisciplinary studies. Therefore, it is recommended to provide brief explanations of some medical terms (define on 1st use), especially when they are first mentioned. RESPONSE: We have defined some of the more common terms, including coeliac disease, intraepithelial lymphocytosis and HLA type in the text. We have added a supplementary table (S2) to define the remaining terms (particularly those in Tables 1, 2 and 5, and Figure 2), so as not to interrupt the flow of the text. In table 6, we have added an additional column “likely significance of test” to make this work more accessible to cross-disciplinary researchers who are not medically qualified. We have also replaced the abbreviation CeD with coeliac disease throughout to make it clearer.
Reviewer 2 Report
Comments and Suggestions for Authors
In this study, the authors have applied a simple audit approach to a large cohort of duodenal biopsies, and the results show that a system capable of identifying most biopsies with normal mucosa and those with features of coeliac disease might achieve automated diagnosis of around 80% of duodenal biopsies. However, The paper is interesting, some minor recommendation for Improvement should be addressed.
- It is recommend that the author explicitly clarify the main contribution of the paper that will be clearly highlighted to help readers understand the significance of the research.
- The manuscript presents a good data auditing about Determination of the relative frequencies of expected diagnoses in duodenal biopsies, but it is lack detailed discussion of how the AI models can benefits from these data to automate this process.
- In page 7, table 2. The authors presented Keywords associated with certain abnormal, non-coeliac diagnostic entities. It is highly recommended to add also if possible the biomarkers findings related to abnormal, non-coeliac diagnostic entities. This will enable the reader to get a complete picture and a big figure about the duodenal biopsies diseases.
- It is recommend that the authors to add a separate section for conclusion and detailed discussion of future research directions.
Author Response
GENERAL RESPONSE: We thank this reviewer for working carefully through our manuscript and providing constructive and helpful suggestions, all of which we have implemented, overall substantially improving the clarity and readability of our submission.
In this study, the authors have applied a simple audit approach to a large cohort of duodenal biopsies, and the results show that a system capable of identifying most biopsies with normal mucosa and those with features of coeliac disease might achieve automated diagnosis of around 80% of duodenal biopsies. However, The paper is interesting, some minor recommendation for Improvement should be addressed.
- It is recommend that the author explicitly clarify the main contribution of the paper that will be clearly highlighted to help readers understand the significance of the research. RESPONSE: We have rewritten parts of the introduction to achieve this.
- The manuscript presents a good data auditing about Determination of the relative frequencies of expected diagnoses in duodenal biopsies, but it is lack detailed discussion of how the AI models can benefits from these data to automate this process. RESPONSE: We have added sentences on this into the introduction and discussion.
- In page 7, table 2. The authors presented Keywords associated with certain abnormal, non-coeliac diagnostic entities. It is highly recommended to add also if possible the biomarkers findings related to abnormal, non-coeliac diagnostic entities This will enable the reader to get a complete picture and a big figure about the duodenal biopsies diseases. RESPONSE: We have made clear in the introduction and the table 2 legend that diagnosis was made on the basis of morphological examination of the biopsy samples, and that there are no specific biomarkers for making these diagnoses. We have also added some explanation to table 6, which includes blood result data, which might otherwise be difficult to interpret for those not from a medical background.
- It is recommend that the authors to add a separate section for conclusion and detailed discussion of future research directions. RESPONSE: We have done as suggested.
Reviewer 3 Report
Comments and Suggestions for Authors
Review Report for MDPI Diagnostics
(Determination of the relative frequencies of expected diagnoses in duodenal biopsies: an essential step in developing an artificial intelligence approach to diagnostic classification)
1. This study presents an 18-month audit of duodenal biopsies to understand the current diagnostic landscape for coeliac disease, revealing that the majority of samples were either normal or within the coeliac spectrum, suggesting that automated systems could reduce the diagnostic workload by up to 80%.
2. In the introduction, the subject of the study, pathologists, endoscopic duodenal biopsies, and artificial intelligence related to digital pathology are mentioned. The literature review of the study needs to be detailed. It is suggested that a literature table consisting of columns such as "dataset, originality, results" be added to this section, especially for studies related to artificial intelligence.
3. When the dataset, amount, and type used in the study are examined in detail within the scope of the study, it is observed that they are at a sufficient level. The collection of the dataset specific to the study increased the quality of the study.
4. The methods section of the study needs to be detailed. In the study where the artificial intelligence-based tool is specified, it is suggested that the artificial intelligence section be expressed more clearly.
5. The results obtained in the study are at an appropriate level when compared to the literature. However, it is recommended that the results be compared to the literature in more detail.
In conclusion, the study is of a certain quality. However, it is absolutely necessary to pay attention to all the sections mentioned above.
Author Response
GENERAL RESPONSE: We thank this reviewer for working carefully through our manuscript and providing constructive and helpful suggestions, almost all of which we have implemented, overall substantially improving the clarity and readability of our submission. We are also grateful for the particularly supportive comments.
Review Report for MDPI Diagnostics
(Determination of the relative frequencies of expected diagnoses in duodenal biopsies: an essential step in developing an artificial intelligence approach to diagnostic classification)
- This study presents an 18-month audit of duodenal biopsies to understand the current diagnostic landscape for coeliac disease, revealing that the majority of samples were either normal or within the coeliac spectrum, suggesting that automated systems could reduce the diagnostic workload by up to 80%. RESPONSE: Thank you. No response required.
- In the introduction, the subject of the study, pathologists, endoscopic duodenal biopsies, and artificial intelligence related to digital pathology are mentioned. The literature review of the study needs to be detailed. It is suggested that a literature table consisting of columns such as "dataset, originality, results" be added to this section, especially for studies related to artificial intelligence. RESPONSE: We recognise that we were not clear enough. Currently no AI is being used for duodenal biopsies and we did not set out to review any literature on this. We simply intended to say that, as we are developing an AI approach to reading duodenal biopsies, we needed to audit what occurs presently. We do not think this requires a literature review table, but we have added additional paragraphs to the introduction to make it much clearer what we set out to do and we hope that this greatly improves the clarity and readability.
- When the dataset, amount, and type used in the study are examined in detail within the scope of the study, it is observed that they are at a sufficient level. The collection of the dataset specific to the study increased the quality of the study. RESPONSE: Thank you. No response required.
- The methods section of the study needs to be detailed. In the study where the artificial intelligence-based tool is specified, it is suggested that the artificial intelligence section be expressed more clearly. RESPONSE: Artificial intelligence was not employed in this study and so it is not mentioned in the methods. We have made it clearer in the introduction and discussion sections how the results of this study could be used in developing AI software-based solutions to reporting duodenal biopsies.
- The results obtained in the study are at an appropriate level when compared to the literature. However, it is recommended that the results be compared to the literature in more detail. RESPONSE: No one has previously audited the diagnoses made in a large series of duodenal biopsies. Neither has anyone investigated the likelihood of a coeliac disease diagnosis in biopsies showing isolated intraepithelial lymphocytosis. Therefore, there is no literature to compare our work to. We have reinforced the fact that no published study tackles either of these questions, both in the introduction and in the discussion. Table 6 already contains a literature review of conditions other than coeliac disease that are associated with isolated intraepithelial lymphocytosis.
In conclusion, the study is of a certain quality. However, it is absolutely necessary to pay attention to all the sections mentioned above. RESPONSE: Thank you for these supportive comments.
Reviewer 4 Report
Comments and Suggestions for Authors
This study audits 18 months’ worth of duodenal biopsies at a tertiary center to better understand the distribution of pathological findings, with a focus on informing the future development of AI diagnostic tools. Among 6,245 biopsies reviewed, the majority were normal (73.76%), with 8.84% falling within the coeliac spectrum and a range of other diagnoses represented in smaller proportions. The authors suggest that these data highlight the potential of automated systems to alleviate diagnostic burden by reliably identifying normal or coeliac-associated biopsies.
While the objective of the paper is clear and the premise clinically relevant—especially in the context of workload management and AI tool development—the Introduction is presented in disjointed subheadings. This format impairs the overall flow of the text. I would strongly recommend rewriting it into a unified, logically structured narrative to improve readability and cohesion.
The Methods section is too short and lacks critical detail. As it stands, it does not provide sufficient clarity on how diagnoses were defined, whether re-review was conducted, what inclusion/exclusion criteria were used, or how interobserver variability was handled. Without this information, the study cannot be considered reproducible, which is a key concern, particularly in papers that aim to serve as the groundwork for computational model development.
In the Results section, there is a tendency to introduce methodological clarifications while presenting data—these should have been fully described earlier. The Discussion is minimal and doesn't adequately reflect on the broader implications, limitations, or next steps. The Conclusion feels somewhat hollow, lacking a synthesis of findings or a strong final message.
Comments on the Quality of English LanguageNeeds improvement.
Author Response
GENERAL RESPONSE: We thank this reviewer for working carefully through our manuscript and providing constructive and helpful suggestions, all of which we have implemented, overall substantially improving the clarity and readability of our submission.
This study audits 18 months’ worth of duodenal biopsies at a tertiary center to better understand the distribution of pathological findings, with a focus on informing the future development of AI diagnostic tools. Among 6,245 biopsies reviewed, the majority were normal (73.76%), with 8.84% falling within the coeliac spectrum and a range of other diagnoses represented in smaller proportions. The authors suggest that these data highlight the potential of automated systems to alleviate diagnostic burden by reliably identifying normal or coeliac-associated biopsies.
While the objective of the paper is clear and the premise clinically relevant—especially in the context of workload management and AI tool development—the Introduction is presented in disjointed subheadings. This format impairs the overall flow of the text. I would strongly recommend rewriting it into a unified, logically structured narrative to improve readability and cohesion. RESPONSE: The introduction has been substantially rewritten, with reordering of some parts of it, to take account of this concern.
The Methods section is too short and lacks critical detail. As it stands, it does not provide sufficient clarity on how diagnoses were defined, whether re-review was conducted, what inclusion/exclusion criteria were used, or how interobserver variability was handled. Without this information, the study cannot be considered reproducible, which is a key concern, particularly in papers that aim to serve as the groundwork for computational model development. RESPONSE: We have substantially expanded the methods section. We have added information on how diagnoses were defined, whether re-review was conducted, what inclusion/exclusion criteria were used, and how interobserver variability was handled. We have added a schematic, figure 1, to improve the clarity of the methods section.
In the Results section, there is a tendency to introduce methodological clarifications while presenting data—these should have been fully described earlier. The Discussion is minimal and doesn't adequately reflect on the broader implications, limitations, or next steps. The Conclusion feels somewhat hollow, lacking a synthesis of findings or a strong final message. RESPONSE: We have moved material from the results to the methods section, as suggested. We have strengthened the results discussion and conclusion sections, as suggested.
Round 2
Reviewer 1 Report
Comments and Suggestions for Authors
The necessary corrections have been made to the article. I recommend accepting the article as it is.
Reviewer 4 Report
Comments and Suggestions for Authors
My concerns have been addressed.